# Behavior–biological mismatch in metabolic health: Evidence from South Korean adults before, during, and after COVID-19 (KNHANES 2019–2022)

Yoo Mi Jeong[1], Minjeong Kim[2], Jae Yeon Jeong[3]*

1 College of Nursing, Dankook University, Dandae-ro Cheonan-si, Chungnam-do, Republic of Korea,
2 San Diego State University, Campanile Dr., San Diego, United states of America, 3 Department of Health Management, Jeonju University, Jeonju-si, Jeollabuk-do, Republic of Korea

* jjy1122@jj.ac.kr

## Abstract

### Objective

Regular physical activity (PA) is recommended for cardiometabolic prevention, yet emerging evidence suggests that meeting PA guidelines alone does not guarantee metabolic health. Pandemic-related disruptions may have intensified behavior–biology discrepancies. This study examined the prevalence and sociodemographic determinants of the physically active but metabolically unhealthy (PA–NH) group before, during, and after the COVID-19 pandemic in Korea.

### Methods

We analyzed 17,719 adults aged ≥19 years using the 2019–2022 Korea National Health and Nutrition Examination Survey. Participants were classified into four groups: PA–H, NPA–H, NPA–NH, and PA–NH. Temporal patterns across the pre- (2019), during- (2020–2021), and post-pandemic period (2022) were assessed using $\chi^2$ tests and ANOVA. Multinomial logistic regression identified determinants of PA–NH membership.

### Results

From 2019 to 2022, PA–H declined during the pandemic and rebounded thereafter, whereas PA–NH increased from 31.2% to 34.0%. Older adults had markedly higher odds of PA–NH (OR 7.30 for ≥70 y vs 19–29 y). Women were less likely to belong to PA–NH. Higher education reduced the odds of mismatch relative to PA–H but increased them compared with NPA–NH. Metabolic risk factors such as hypertension and hypercholesterolemia rose despite recovery in PA.

**Data availability statement:** All the data files are available from the KNHANES database (website: https://chs.kdca.go.kr/).

**Funding:** The author(s) received no specific funding for this work.

**Competing interests:** he authors have declared that no competing interests exist.

## Conclusions

The persistence and rise of PA–NH after the pandemic show an association suggesting that adherence to PA guidelines alone is insufficient for cardiometabolic protection. Post-pandemic public health strategies should integrate aerobic and resistance exercise with dietary modification, stress management, and routine screening, prioritizing older adults and other high-risk groups.

## Introduction

Regular physical activity (PA) is a central behavioral determinant of cardiometabolic health, with extensive evidence demonstrating its benefits for blood pressure, glucose regulation, lipid metabolism, and obesity prevention [1–3]. However, accumulating research indicates that adherence to PA guidelines alone does not uniformly translate into favorable biological outcomes. A subgroup of adults exhibits metabolic abnormalities despite meeting recommended PA levels—often described as a behavior–biological mismatch [4,5]. Although this pattern has been documented in multiple cohorts, its temporal trajectory and sociodemographic distribution remain insufficiently understood, particularly in the context of large-scale disruptions such as the COVID-19 pandemic.

The COVID-19 pandemic substantially altered daily routines and health behaviors. Restrictions on mobility and the closure of facilities reduced opportunities for structured PA [6,7], while psychosocial stress, dietary changes, and delays in healthcare utilization may have aggravated metabolic risk [5,8]. Korea experienced relatively shorter and less stringent restrictions compared with many Western countries [9,10], providing a unique context in which to examine whether discrepancies between behavior and metabolic outcomes persisted or intensified during and after the pandemic period.

Understanding the evolution and determinants of the physically active but metabolically unhealthy (PA–NH) group is essential, as their existence suggests that behavioral compliance alone may not ensure metabolic resilience [4,5]. Therefore, using nationally representative data, this study examined temporal changes and sociodemographic correlates of the PA–NH phenotype before (2019), during (2020–2021), and after (2022) the COVID-19 pandemic in South Korea.

## Methods

### 1. Sample and data collection

This cross-sectional study used data from the 2019–2022 Korea National Health and Nutrition Examination Survey (KNHANES), a nationally representative survey conducted annually by the Korea Disease Control and Prevention Agency using multistage stratified cluster sampling [7]. To reflect the pandemic context, years were grouped as pre-pandemic (2019), during-pandemic (2020–2021), and post-pandemic (2022).

Among 23,356 adults aged ≥19 years with available interview and examination data, we excluded cases with missing values for PA or metabolic indicators, yielding a final analytic sample of 17,719 participants (4,796 in 2019; 4,260 in 2020; 4,458 in 2021; and 4,205 in 2022).

## 2. Ethics and data gathering procedure

The data supporting this study is publicly available secondary data, specifically the de-identified dataset from the Korea National Health and Nutrition Examination Survey (KNHANES). This dataset is anonymized, ensuring the prevention of individual participant identification. The data was accessed on April 24, 2024, via the Korea Disease Control and Prevention Agency(KDCA) website (https://chs.kdca.go.kr/). This analysis used publicly available de-identified secondary data and received institutional approval from D University IRB (DKU 2025-01-010-003).

### Measures

**Physical activity (PA).** PA status was defined based on KDCA criteria aligned with WHO guidelines [11,12]. PA was defined as meeting aerobic, walking, or muscle-strengthening guidelines., at least 75 minutes of vigorous-intensity activity, or an equivalent combination of both. Aerobic PA was defined as having engaged in at least 150 minutes of moderate-intensity activity per week, at least 75 minutes of vigorous-intensity activity, or an equivalent combination of both. Walking was defined as having walked for at least 10 minutes per session, totaling 30 minutes or more per day for at least five days during the past week. Muscle-strengthening activity was defined as having performed exercises such as push-ups, sit-ups, dumbbell or barbell training, or weightlifting for at least two days during the past week. Participants meeting ≥1 of these three guidelines were assigned a value of 1, otherwise 0.

**Metabolic risk.** Metabolic risk was defined as the presence of ≥1 of the following conditions: diabetes, hypertension, hypercholesterolemia, hypertriglyceridemia, or obesity. Metabolic syndrome was defined as having ≥3 conditions [2,13]. Diabetes was defined as having a fasting blood glucose level ≥126 mg/dL, a previous diagnosis by a physician, the use of glucose-lowering medication or insulin, or a hemoglobin A1c level ≥6.5%. Hypertension was defined as a systolic blood pressure ≥140 mmHg, a diastolic blood pressure ≥90 mmHg, or the current use of antihypertensive medication. Hypercholesterolemia was defined as having a total cholesterol level ≥240 mg/dL or the use of cholesterol-lowering medication. Hypertriglyceridemia was defined as a triglyceride level ≥200 mg/dL. Obesity was defined as a body mass index (BMI) ≥25 kg/m², on the basis of the Asia–Pacific WHO criteria.

**Behavior–biological classification.** Participants were categorized into four behavior–biological groups on the basis of their engagement in physical activity and the presence of metabolic risk factors. The groups were defined as follows: PA-H: Participants who engaged in recommended physical activity and had no metabolic risk factors. NPA-H: Participants who did not engage in physical activity but had no metabolic risk factors. NPA-NH: Participants who did not engage in physical activity and had at least one metabolic risk factor. PA-NH: Participants who engaged in physical activity but had at least one metabolic risk factor. This classification was used to assess changes in the combined patterns of health behavior and metabolic risk across different phases of the COVID-19 pandemic.

**Covariates.** Covariates included sex, age (19–29, 30–39, 40–49, 50–59, 60–69, ≥70 years), household income (quartiles), education (≤elementary, middle, high school, ≥college), and employment status (employed vs unemployed) [13].

### Statistical analysis

All analyses accounted for the complex survey design using Stata 18.5MP (StataCorp, College Station, TX). Trends in behavior–biological groups and metabolic indicators across the three periods were assessed using $\chi^2$ tests and ANOVA. Multinomial logistic regression models estimated odds ratios (ORs) and 95% confidence intervals (CIs) for membership in PA–NH relative to the three comparison groups (PA–H, NPA–H, NPA–NH). Two-sided $p < 0.05$ was considered statistically significant.

## Results

### 1. Sociodemographic characteristics

Table 1 summarizes participant characteristics across survey years. Sex distribution remained stable ($p = 0.531$). The age profile shifted, with a higher proportion of adults aged ≥60 years over time ($p < 0.001$). Education level increased modestly

**Table 1. General characteristics of the participants categorized by the respective year.**

| Variable | | 2019(n = 4,796) | | 2020(n = 4,260) | | 2021(n = 4,458) | | 2022(n = 4,205) | | *P*-value |
|---|---|---|---|---|---|---|---|---|---|---|
| Sex | Male | 2,072 | 43.2% | 1,887 | 44.3% | 1,947 | 43.7% | 1,799 | 42.8% | 0.531 |
| | Female | 2,724 | 56.8% | 2,373 | 55.7% | 2,511 | 56.3% | 2,406 | 57.2% | |
| Age | 19-29 | 544 | 11.3% | 588 | 13.8% | 545 | 12.2% | 493 | 11.7% | 0.000 |
| | 30-39 | 699 | 14.6% | 579 | 13.6% | 506 | 11.4% | 514 | 12.2% | |
| | 40-49 | 876 | 18.3% | 708 | 16.6% | 783 | 17.6% | 697 | 16.6% | |
| | 50-59 | 911 | 19.0% | 803 | 18.9% | 824 | 18.5% | 736 | 17.5% | |
| | 60-69 | 890 | 18.6% | 844 | 19.8% | 908 | 20.4% | 948 | 22.5% | |
| | ≥ 70 | 876 | 18.3% | 738 | 17.3% | 892 | 20.0% | 817 | 19.4% | |
| Income | Q1 | 943 | 19.7% | 819 | 19.2% | 867 | 19.5% | 825 | 19.6% | 1.000 |
| | Q2 | 936 | 19.5% | 847 | 19.9% | 884 | 19.8% | 828 | 19.7% | |
| | Q3 | 978 | 20.4% | 859 | 20.2% | 902 | 20.2% | 836 | 19.9% | |
| | Q4 | 974 | 20.3% | 858 | 20.1% | 894 | 20.1% | 851 | 20.2% | |
| | Q5 | 965 | 20.1% | 877 | 20.6% | 911 | 20.4% | 865 | 20.6% | |
| Education | ≤ Elementary school | 890 | 18.6% | 713 | 16.7% | 835 | 18.7% | 760 | 18.1% | 0.012 |
| | Middle school | 459 | 9.6% | 407 | 9.6% | 449 | 10.1% | 352 | 8.4% | |
| | High school | 1,586 | 33.1% | 1,490 | 35.0% | 1,482 | 33.2% | 1,382 | 32.9% | |
| | ≥ College | 1,861 | 38.8% | 1,650 | 38.7% | 1,692 | 38.0% | 1,711 | 40.7% | |
| Employment status | Employed | 2,846 | 59.3% | 2,471 | 58.0% | 2,664 | 59.8% | 2,563 | 61.0% | 0.050 |
| | Unemployed | 1,950 | 40.7% | 1,789 | 42.0% | 1,794 | 40.2% | 1,642 | 39.1% | |

n, %, the χ2 test has been performed

(p = 0.012), while employment remained more prevalent than unemployment across all periods with borderline temporal variation (p = 0.050). Household income distribution was stable (p = 1.000).

## 2. Trends in physical activity and metabolic risk

As shown in Table 2, the proportion of participants meeting ≥1 PA recommendation increased from 51.5% in 2019 to 55.0% in 2022 (p = 0.005). Engagement in aerobic PA, walking, and muscle-strengthening PA increased significantly across the study period. Metabolic indicators exhibited heterogeneous trends. Diabetes and obesity modestly decreased (p = 0.017 and p = 0.002, respectively), whereas hypertension and hypercholesterolemia increased (both p < 0.001). Hyper-triglyceridemia showed no significant change (p = 0.465). The prevalence of ≥1 metabolic risk factor increased steadily (p < 0.001), and metabolic syndrome rose during the pandemic and did not return to pre-pandemic levels.

## 3. Behavior–biological group changes

Table 3 shows the distribution of participants according to the behavior–biological group classification. The proportion of the PA-H group decreased during the COVID-19 period but rebounded to 21.0% in 2022. The NPA-H group declined steadily from 16.3% in 2019 to 12.5% in 2022. The NPA-NH group consistently accounted for the largest share (32.1%–33.4%) without a clear temporal trend. The PA-NH group slightly increased over time, from 31.2% in 2019 to 34.0% in 2022 (p < 0.001).

## 4. Factors associated with PA–NH

Multinomial logistic regression results are presented in Table 4. Women were significantly less likely than men to belong to the PA–NH group (p < 0.001). Older age was strongly associated with higher odds of PA–NH relative to the PA–H and

**Table 2. Health behavior and metabolic risk of the participants categorized by the respective year.**

| Variable | | 2019(n = 4,796) | | 2020(n = 4,260) | | 2021(n = 4,458) | | 2022(n = 4,205) | | P-value |
|---|---|---|---|---|---|---|---|---|---|---|
| Aerobic Physical Activity | Yes | 2,055 | 42.8% | 1,797 | 42.2% | 1,870 | 42.0% | 1,935 | 46.0% | 0.000 |
| | No | 2,741 | 57.2% | 2,463 | 57.8% | 2,588 | 58.0% | 2,270 | 54.0% | |
| Walking exercise | Yes | 145 | 3.0% | 143 | 3.4% | 206 | 4.6% | 168 | 4.0% | 0.000 |
| | No | 4,651 | 97.0% | 4,117 | 96.6% | 4,252 | 95.4% | 4,037 | 96.0% | |
| Muscle-strengthening Exercise | Yes | 1,030 | 21.5% | 1,007 | 23.6% | 1,047 | 23.5% | 1,000 | 23.8% | 0.026 |
| | No | 3,766 | 78.5% | 3,253 | 76.4% | 3,411 | 76.5% | 3,205 | 76.2% | |
| Health behavior* | Yes | 2,471 | 51.5% | 2,211 | 51.9% | 2,347 | 52.6% | 2,313 | 55.0% | 0.005 |
| | No | 2,325 | 48.5% | 2,049 | 48.1% | 2,111 | 47.4% | 1,892 | 45.0% | |
| Diabetes | Yes | 699 | 14.6% | 674 | 15.8% | 728 | 16.3% | 598 | 14.2% | 0.017 |
| | No | 4,097 | 85.4% | 3,586 | 84.2% | 3,730 | 83.7% | 3,607 | 85.8% | |
| Hypertension | Yes | 1,575 | 32.8% | 1,367 | 32.1% | 1,479 | 33.2% | 1,424 | 33.9% | 0.000 |
| | No | 3,221 | 67.2% | 2,893 | 67.9% | 2,979 | 66.8% | 2,781 | 66.1% | |
| Hypercholesterolemia | Yes | 1,216 | 25.3% | 1,136 | 26.7% | 1,297 | 29.1% | 1,301 | 30.9% | 0.000 |
| | No | 3,580 | 74.7% | 3,124 | 73.3% | 3,161 | 70.9% | 2,904 | 69.1% | |
| Hypertriglyceridemia | Yes | 612 | 12.8% | 557 | 13.1% | 536 | 12.0% | 519 | 12.3% | 0.465 |
| | No | 4,184 | 87.2% | 3,703 | 86.9% | 3,922 | 88.0% | 3,686 | 87.7% | |
| Obesity | Yes | 1,637 | 34.1% | 1,622 | 38.1% | 1,616 | 36.2% | 1,507 | 35.8% | 0.002 |
| | No | 3,159 | 65.9% | 2,638 | 61.9% | 2,842 | 63.8% | 2,698 | 64.2% | |
| Metabolic risk | No | 1,754 | 36.6% | 1,452 | 34.1% | 1,498 | 33.6% | 1,410 | 33.5% | 0.000 |
| | Metabolic risk | 2,278 | 47.5% | 2,075 | 48.7% | 2,182 | 48.9% | 2,080 | 49.5% | |
| | Metabolic syndrome | 764 | 15.9% | 733 | 17.2% | 778 | 17.5% | 715 | 17.0% | |

n, %, the χ2 test has been performed

Health behavior: coded as 1 if the participant met at least one of the three recommended physical activity guidelines.

Metabolic risk defined as having ≥1 of following diabetes, hypertension, hypercholesterolemia, hypertriglyceridemia, or obesity; metabolic syndrome ≥3 of these conditions.

**Table 3. Behavior-biological group changes by the respective year.**

| Variable | | 2019(n = 5,875) | | 2020(n = 5,379) | | 2021(n = 5,277) | | 2022(n = 4,844) | | P-value |
|---|---|---|---|---|---|---|---|---|---|---|
| Behavior-Biological Group | PA-H | 974 | 20.3% | 772 | 18.1% | 874 | 19.6% | 884 | 21.0% | 0.000 |
| | NPA-H | 780 | 16.3% | 680 | 16.0% | 624 | 14.0% | 526 | 12.5% | |
| | NPA-NH | 1,545 | 32.2% | 1,369 | 32.1% | 1,487 | 33.4% | 1,366 | 32.5% | |
| | PA-NH | 1,497 | 31.2% | 1,439 | 33.8% | 1,473 | 33.0% | 1,429 | 34.0% | |

n (%), the χ2 test and ANOVA has been performed

PA-H: Participants who engaged in recommended physical activity and had no metabolic risk factors.

NPA-H: Participants who did not engage in physical activity but had no metabolic risk factors.

NPA-NH: Participants who did not engage in physical activity and had at least one metabolic risk factor.

PA-NH: Participants who engaged in physical activity but had at least one metabolic risk factor.

NPA–H groups, with the highest estimates observed among adults aged ≥70 years (OR 7.30 and OR 4.76, respectively; both p < 0.001). In contrast, compared with the NPA–NH group, all older age categories showed significantly lower odds of PA–NH relative to adults aged 19–29 years. Household income and employment status were not significantly associated with PA–NH in most models. Education demonstrated a distinct bidirectional pattern: higher educational attainment was

**Table 4. Factors associated with belonging to the behavior–biological mismatch group.**

| | | PA-NH(ref.PA-H) | | PA-NH(ref.NPA-H) | | PA-NH(ref.NPA-NH) | |
|---|---|---|---|---|---|---|---|
| | | OR(S.E) | *p* | OR(S.E) | *p* | OR(S.E) | *p* |
| Sex(ref.Men) | Women | 0.47(0.02) | 0.000 | 0.30(0.02) | 0.000 | 0.66(0.03) | 0.000 |
| Age(ref.19–29) | 30-39 | 1.92(0.16) | 0.000 | 1.04(0.09) | 0.664 | 0.64(0.07) | 0.000 |
| | 40-49 | 2.70(0.21) | 0.000 | 1.49(0.13) | 0.000 | 0.51(0.05) | 0.000 |
| | 50-59 | 4.49(0.35) | 0.000 | 3.11(0.28) | 0.000 | 0.56(0.05) | 0.000 |
| | 60-69 | 6.34(0.54) | 0.000 | 5.29(0.53) | 0.000 | 0.63(0.06) | 0.000 |
| | ≥ 70 | 7.30(0.78) | 0.000 | 4.76(0.56) | 0.000 | 0.47(0.05) | 0.000 |
| Income(ref.Q1) | Q2 | 0.91(0.07) | 0.190 | 0.99(0.08) | 0.904 | 0.99(0.06) | 0.816 |
| | Q3 | 0.87(0.06) | 0.061 | 0.94(0.07) | 0.422 | 1.05(0.06) | 0.427 |
| | Q4 | 0.89(0.07) | 0.117 | 1.00(0.08) | 0.997 | 1.08(0.07) | 0.216 |
| | Q5 | 0.86(0.06) | 0.039 | 1.08(0.09) | 0.318 | 1.14(0.07) | 0.039 |
| Education | Middle school | 0.76(0.09) | 0.024 | 1.21(0.15) | 0.120 | 1.37(0.10) | 0.000 |
| (ref. ≤ elementary school) | High school | 0.55(0.06) | 0.000 | 1.04(0.11) | 0.698 | 1.66(0.10) | 0.000 |
| | ≥ College | 0.49(0.05) | 0.000 | 1.07(0.11) | 0.530 | 2.17(0.15) | 0.000 |
| Employment status (ref. employed) | Unemployed | 1.05(0.05) | 0.341 | 1.05(0.06) | 0.326 | 1.05(0.04) | 0.283 |

associated with lower odds of PA–NH compared with PA–H; no significant associations were observed when compared with NPA–H; and higher odds of PA–NH were observed when compared with NPA–NH (p < 0.001).

## Discussion

This study examined temporal changes in physical activity (PA) and metabolic risk among Korean adults from 2019 to 2022, focusing on the physically active but metabolically unhealthy (PA–NH) group. Despite partial recovery in PA participation after the relaxation of COVID-19 restrictions, the prevalence of PA–NH increased and persisted. These findings suggest associations indicating that adherence to PA recommendations alone may not ensure cardiometabolic resilience.

PA participation increased modestly overall, and the proportion of the PA–H group partially recovered in 2022. This pattern contrasts with reports from other countries showing sustained declines in PA during prolonged lockdowns [11,14] but aligns with Korean surveillance data demonstrating gradual behavioral rebound [7]. Korea's relatively shorter and less stringent restrictions [9,10] and the widespread transition to walking or home-based exercise, may have mitigated prolonged reductions in PA. However, the steady decline in the NPA–H group indicates that prolonged inactivity may erode metabolic resilience, consistent with evidence that sedentary behavior exerts cumulative metabolic harm [15].

Trends in metabolic indicators were heterogeneous. While obesity and diabetes slightly decreased, hypertension and hypercholesterolemia increased and did not return to pre-pandemic levels, consistent with prior Korean studies [16,17]. Pandemic-related stress, dietary changes, altered sleep patterns, and delayed healthcare access may have contributed to these adverse trends independent of PA [5,8]. These patterns support the interpretation that PA alone may be insufficient to counteract broader physiological and behavioral stressors affecting metabolic regulation.

The rise in PA–NH highlights a subgroup exhibiting biological vulnerability despite behavioral adherence. Older adults were overrepresented in PA–NH, likely reflecting age-related declines in skeletal muscle mass, insulin sensitivity, and vascular function [18]. The relative overrepresentation of younger adults in PA–NH compared with NPA–NH suggests that pandemic-related metabolic susceptibility may persist even when active lifestyles resume. Educational attainment showed a bidirectional association: higher education decreased PA–NH odds relative to PA–H but increased PA–NH odds relative to NPA–NH, potentially reflecting prolonged sedentary time and occupational stress among highly educated individuals.

Income and employment showed no consistent associations, suggesting that mismatch is not confined to socioeconomic disadvantage when PA status is considered [13].

These findings underscore the need for integrated metabolic prevention strategies that combine aerobic and resistance training with dietary improvement, stress reduction, and routine metabolic screening [16,19]. Older adults and individuals exposed to sustained occupational or educational stress may particularly benefit from targeted interventions.

Strengths of this study include the use of nationally representative data across pre-pan- demic, pandemic, and post-pandemic periods; its large sample enabling subgroup analysis; and biomarker-based metabolic indicators reducing self-report bias. Limitations include the cross-sectional design, reliance on self-reported PA, and the absence of dietary, sleep, medication, or genetic data. PA classification relied solely on WHO-aligned aerobic thresholds, and institutionalized adults were not included, potentially limiting generalizability.

## Conclusion

Using nationally representative data spanning the pre-, during-, and post-pandemic periods, this study showed that the prevalence of individuals who are physically active but metabolically unhealthy (PA–NH) increased and persisted after COVID-19. These findings suggest associations indicating that adherence to PA guidelines alone may not ensure car-diometabolic protection, particularly among older adults and individuals exposed to complex lifestyle or stress-related factors. Prevention efforts should extend beyond aerobic PA to incorporate resistance training, dietary management, stress reduction, and routine metabolic screening. Targeting high-risk subgroups with integrated, multimodal strategies may be essential for restoring metabolic resilience in the post-pandemic context.

## Author contributions

**Data curation:** Jae Yeon Jeong.

**Formal analysis:** Jae Yeon Jeong.

**Investigation:** Yoo Mi Jeong, Minjeong Kim.

**Methodology:** Jae Yeon Jeong.

**Project administration:** Yoo Mi Jeong.

**Software:** Jae Yeon Jeong.

**Supervision:** Yoo Mi Jeong, Minjeong Kim.

**Validation:** Yoo Mi Jeong, Minjeong Kim, Jae Yeon Jeong.

**Visualization:** Jae Yeon Jeong.

**Writing – original draft:** Yoo Mi Jeong, Jae Yeon Jeong.

**Writing – review & editing:** Yoo Mi Jeong, Minjeong Kim.

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
