## [Decision Letter · Decision Letter 0]

20 Feb 2026

Dear Dr. Jeong,

Thank you for submitting your manuscript to PLOS ONE. After careful consideration, we feel that it has merit but does not fully meet PLOS ONE’s publication criteria as it currently stands. Therefore, we invite you to submit a revised version of the manuscript that addresses the points raised during the review process.

We look forward to receiving your revised manuscript.

Kind regards,

Kwang-Sig Lee

Academic Editor

PLOS One

Journal Requirements:

https://journals.plos.org/plosone/s/file?id =. wjVg/PLOSOne_formatting_sample_main_body.pdf and

Reviewers' comments:

Reviewer's Responses to Questions

**Comments to the Author**

1. Is the manuscript technically sound, and do the data support the conclusions?

Reviewer #1: Yes

2. Has the statistical analysis been performed appropriately and rigorously?

Reviewer #1: Yes

3. Have the authors made all data underlying the findings in their manuscript fully available?

Reviewer #1: Yes

4. Is the manuscript presented in an intelligible fashion and written in standard English?

Reviewer #1: Yes

Reviewer #1: This manuscript addresses an important and timely topic with potential relevance to both clinical practice and public health. The study is generally well structured and presents its findings in a clear and logical manner. The research question is relevant, and the authors have provided sufficient background to justify the study. However, few issues should be addressed to improve the rigor and clarity of the manuscript.

The authors conclude that PA alone is insufficient for cardiometabolic protection. This conclusion is supported, but causal language should be avoided due to the cross-sectional design.I would recommend adding: “associations” instead of “indicate” or “demonstrate.”

Also, there are minor grammatical errors—e.g., inconsistent tense in PA classification description.

**Do you want your identity to be public for this peer review?** For information about this choice, including consent withdrawal, please see our Privacy Policy

Reviewer #1: **Yes:** Chibuzor Stella Amadi

---

## [Author Response · Author response to Decision Letter 1]

23 Feb 2026

Thank you for the suggestion. We agree that given the cross-sectional nature of the study, it is more appropriate to use language that emphasizes associations rather than direct causality. In accordance with this suggestion, we have replaced causal terms such as "indicate" and "demonstrate" with "show an association suggesting" or "suggest associations indicating" throughout the manuscript, particularly in the Abstract, Discussion, and Conclusion sections.

We have thoroughly reviewed the "Physical activity (PA)" section in the Methods and ensured that the tense in the PA classification descriptions is consistent. Specifically, we have corrected the instances where the present tense was used and unified them into the past tense to describe the study procedures.

We have conducted a final proofreading of the manuscript and tables. We have corrected the typo in Table 4 (e.g., changed "ge" to "Age") and the capitalization in Table 1 and Table 4. The numerical data in Table 2 have been consistently adjusted for decimal places(e.g., changed "23.78" to "23.8%", “76.22” to “76.2%”).

---

## [Editor Report · Decision Letter 1]

26 Feb 2026

Behavior–Biological Mismatch in Metabolic Health: Evidence from South Korean Adults Before, During, and After COVID-19 (KNHANES 2019–2022)

PONE-D-25-63798R1

Dear Dr. Jeong,

We’re pleased to inform you that your manuscript has been judged scientifically suitable for publication and will be formally accepted for publication once it meets all outstanding technical requirements.

Kind regards,

Kwang-Sig Lee

Academic Editor

PLOS One
---

## [Editor Report · Acceptance letter]

PONE-D-25-63798R1

PLOS One

Dear Dr. Jeong,

I'm pleased to inform you that your manuscript has been deemed suitable for publication in PLOS One. Congratulations! Your manuscript is now being handed over to our production team.

Kind regards,

on behalf of

Professor Kwang-Sig Lee

Academic Editor

PLOS One